# Coagulation and Fibrinolysis in Obstructive Sleep Apnoea

**DOI:** 10.3390/ijms22062834

**Published:** 2021-03-11

**Authors:** Andras Bikov, Martina Meszaros, Esther Irene Schwarz

**Affiliations:** 1North West Lung Centre, Manchester University NHS Foundation Trust, Manchester M23 9LT, UK; 2Division of Infection, Immunity and Respiratory Medicine, University of Manchester, Manchester M13 9MT, UK; 3Department of Pulmonology, Semmelweis University, 1083 Budapest, Hungary; martina.meszaros@usz.ch; 4Department of Pulmonology and Sleep Disorders Centre, University Hospital Zurich, 8006 Zurich, Switzerland; estherirene.schwarz@usz.ch; 5Centre of Competence Sleep & Health Zurich, University of Zurich, 8091 Zurich, Switzerland

**Keywords:** obstructive sleep apnoea, OSA, coagulation, fibrinolysis, platelets

## Abstract

Obstructive sleep apnoea (OSA) is a common disease which is characterised by repetitive collapse of the upper airways during sleep resulting in chronic intermittent hypoxaemia and frequent microarousals, consequently leading to sympathetic overflow, enhanced oxidative stress, systemic inflammation, and metabolic disturbances. OSA is associated with increased risk for cardiovascular morbidity and mortality, and accelerated coagulation, platelet activation, and impaired fibrinolysis serve the link between OSA and cardiovascular disease. In this article we briefly describe physiological coagulation and fibrinolysis focusing on processes which could be altered in OSA. Then, we discuss how OSA-associated disturbances, such as hypoxaemia, sympathetic system activation, and systemic inflammation, affect these processes. Finally, we critically review the literature on OSA-related changes in markers of coagulation and fibrinolysis, discuss potential reasons for discrepancies, and comment on the clinical implications and future research needs.

## 1. Introduction

Obstructive sleep apnoea (OSA) is a common sleep-related breathing disorder that has been associated with an increased incidence of thromboembolic, cardio- and cerebrovascular events. The main direct pathophysiological consequences of the repetitive collapse of the upper airway during sleep resulting in apnoeas and hypopnoeas are intermittent hypoxaemia, intrathoracic pressure swings, and arousals. Sympathetic overdrive, hypertension, oxidative stress, shear stress, and metabolic derangements promote endothelial dysfunction in OSA [1,2,3,4]. In addition, there is growing awareness that there are different phenotypes of OSA based on pathophysiology, symptoms, comorbidities, and long-term cardiovascular consequences [5,6]. The factors promoting endothelial dysfunction might differ between phenotypes, and findings of mechanistic studies might not be applicable to all patients with OSA.

OSA has also been shown to result in a hypercoagulable state, and hypercoagulability has been proposed as one of the contributing mechanisms for the observed increased risk of vascular events in OSA [7,8]. Several studies have reported that fibrinogen and other prothrombotic factors are increased and that fibrinolytic capacity is reduced in OSA. Endothelial dysfunction, an increase in prothrombotic factors [9], a decrease in fibrinolytic activity, platelet activation, and changed rheology and viscosity (e.g., increase in haematocrit as a consequence of nocturnal hypoxaemia) are potential mechanisms explaining a prothrombotic state in OSA [10]. Although there are several convincing theories for how the pathophysiological consequences of OSA might result in a procoagulant state, either via endothelial dysfunction or through interfering with the coagulation cascade or fibrinolysis, there is only limited quality evidence on a direct causative relationship between OSA and a procoagulant state.

To identify relevant articles in this field, the search engine of PubMed was used. The search strategy was conducted using a combination of keywords in the following order: “coagulation AND sleep apnoea” OR “coagulation AND OSA” OR “fibrinolysis AND OSA” OR “coagulation AND hypoxia” OR “coagulation AND inflammation” OR “fibrinolysis AND hypoxia” OR “fibrinolysis AND inflammation OR “platelet AND sleep apnoea”, OR “platelet and OSA” OR “platelet AND hypoxia” OR “platelet AND inflammation”. Moreover, we used all molecule names that were found in the review, for example “factor XII AND sleep apnoea” OR “factor XII AND OSA” OR “TF AND sleep apnoea” OR “TF AND OSA” OR “fibrinogen AND sleep apnoea” OR “fibrinogen AND OSA” OR “PAI-1 AND sleep apnoea” OR “PAI-1 AND OSA”. Animal and human studies were included. We selected case control, randomised controlled, and interventional studies and also thematic reviews and systematic reviews/meta-analyses. We included articles written in English published prior to 15.01.2021.

Hypotheses and current evidence on how OSA might promote a disturbance of haemostasis and coagulation–fibrinolysis balance which results in a prothrombotic state, data on the effects of OSA treatment on coagulation activity, and the fibrinolytic system as well as knowledge gaps and future perspectives are discussed.

## 2. Overview of the Coagulation System and Fibrinolysis and the Role of Platelets

The coagulation system works by adhesion and aggregation of activated platelets (known as “primary haemostasis”) and formation of a fibrin network via the coagulation cascade (known as “secondary haemostasis”) [11]. This review focuses mainly on the secondary homeostasis and its role in the pathogenesis of OSA.

The coagulation cascade includes a series of proteolytic events in which serine proteases activate proenzymes on the surface of activated platelets. The cascade has traditionally been divided into “intrinsic” and “extrinsic” pathways. Current literature divides the process of coagulation into “initiation”, “amplification”, “propagation”, and “stabilization” phases [11] (Figure 1).

The activation of the tissue factor (TF, also known as factor III) in combination with factor VII has been considered as the first step of the coagulation cascade. TF is a membrane-bound glycoprotein which is constitutively presented in the subendothelium [12]. In case of endothelial injury, TF is exposed to plasma procoagulants and binds factor VII which is activated (factor VIIa). Moreover, inducible TF can be expressed by inflammatory cells, for example in monocytes in response to endotoxin [13] and by vascular endothelial cells following induction by cytokines such as tumour necrosis factor α (TNFα) [14]. Via calcium signalling, TF–factor VIIa complex binds factor X and catalyses its conversion to factor Xa [15]. Factor Xa can be generated as well by the complex of factor IXa and factor VIIIa after the activation of factor XII–factor XI complex (kallikrein–kinin system, reviewed in detail [16]). Factor Xa binds and activates factor V, and together they form the prothrombinase complex (factor Xa–factor Va–factor II). The prothrombinase complex generates thrombin (factor IIa) from prothrombin (factor II) resulting in the conversion of circulating fibrinogen (factor I) to insoluble fibrin (factor Ia). This complex is the most efficient in the presence of calcium and phospholipid surface of the activated platelets [11]. Finally, fibrin is covalently cross-linked by factor XIII resulting in fibrin polymers which are the major constituents of the clot [17]. Factor XIII binds other anti-fibrinolytic proteins to fibrin such as α2-antiplasmin (A2AP), thrombin activated fibrinolysis inhibitor (TAFI), and complement C3. C3 deposits are associated with thinner fibrin fibres, while they do not affect plasmin formation [18]. Another essential mechanism for clot stability is the presence of activated platelets [19]. Platelets compress and reduce the volume of the thrombus, and they interact with fibrin fibres via the GP IIb–IIIa complex.

The thrombus is lysed by plasmin following activation of plasminogen, which is a zymogen produced by the liver. The activation can occur on the surface of the thrombus or the endothelial cell mainly through the tissue plasminogen activator (tPA) or urokinase (uPA) [20]. The production of both enzymes is induced by thrombin [21]. In plasma the main activator is the tPA as it has higher affinity to plasminogen than uPA, whilst uPA is more important in the extravascular activation of plasmin and is involved in cell migration and wound healing [22]. The tPA is predominantly released by the endothelial cells and requires fibrin as a cofactor, while uPA is produced by monocytes and the urinary epithelium and does not need fibrin for its action. Apart from tPA and uPA, members of the contact pathway, such as activated factor XII, activated factor XI, and kallikrein can also activate plasminogen [23].

Fibrinolysis occurs both on the thrombus and the surface of cells; however, the former process is more effective. The fibrin-bound tPa has significantly higher catalytic activity to activate plasminogen compared to the fluid phase [24]. In addition, the effect of A2AP is inactivated if the plasmin is bound to fibrin [25]. Moreover, both fibrinogen and fibrin facilitate plasmin conversion [26]. Cell-surface-related fibrinolysis is achieved by two main mechanisms. Annexin II may form a complex with S100A10 that binds plasminogen and tPA independently from fibrin [27]. The other mechanism is mediated by the urokinase type plasminogen activator receptor (uPAR) which binds urokinase with plasminogen [28]. The expression and cleavage of uPAR is upregulated by TNFα, interleukin 1β (IL-1β), and IL-6 [28]. The cleavage is also facilitated by uPA and plasmin leading to soluble uPAR (suPAR) [28] which is a proinflammatory molecule but can also act as a scavenger receptor for uPA, inhibiting its function [29].

Once fibrin polymers are degraded by plasmin, fibrin degradation products (FDPs) are formed. Some of these have immunoregulatory and thrombosis modulatory roles. The most commonly used FDP in clinical practice is the d-dimer, which reflects thrombus formation and fibrinolysis [20].

The physiological coagulation cascade is regulated by three main processes. First, antithrombin forms an inhibiting complex with thrombin and factor Xa, called the thrombin–antithrombin (TAT) complex [30]. Second, tissue factor pathway inhibitor (TFPI), which is presented on endothelial cells, inhibits the action of the TF–factor VIIa complex. Third, thrombin forms a complex with endothelial membrane-anchored thrombomodulin and activates protein C. Activated protein C (APC) with its cofactor protein S degrades factors Va and VIIIa resulting in downregulation of the coagulation system [11].

Plasminogen activators are inhibited by plasminogen activator inhibitor-1 (PAI-1), PAI-2, A2AP, α2-macroglobulin, C1-esterase inhibitor, and protease nexin-1. PAI-1 is released by the endothelial cells and platelets and is the most important inhibitor of tPA and uPA [31]. It is upregulated by thrombin; various proinflammatory cytokines, such as TNFα, IL-6, C-reactive protein (CRP), and transforming growth factor beta (TGF-β); as well as hormones, such as insulin and cortisol [32]. α2-antiplasmin is produced by the liver and is a potent inhibitor of plasmin. It regulates fibrinolysis in three ways: by inhibiting adsorption of plasminogen to fibrin, forming complexes with plasmin, and making fibrin more resistant to plasmin through cross-linking with factor XIIIa [33]. It seems that factor XIII is essential to its mechanism [34] as it is inactive if the plasmin is bound to fibrin without XIII [25]. A further mechanism contributing to the regulation of fibrinolysis involves TAFI. This molecule is synthesised in the liver and decreases the number of available plasminogen binding sites, slowing down the fibrinolysis [20].

Activated platelets have a pivotal role in coagulation by providing an activated membrane (such as surface phospholipids) to the coagulation factors and aggregating in the haemostatic plug. Following endothelial injury, platelets are exposed to the highly thrombogenic subendothelium. Subendothelial proteins, such as von Willebrand factor (vWF), collagen, thrombospondin, and vitronectin, bind to several surface glycoprotein (GP) receptors of the platelets [11]. vWF multimers bind the platelet GP–Ib–IX–V complex which reduces platelet velocity. Thus, collagen fibres are able to bind platelet GP–VI and GP–Ia–IIa complexes, anchoring the platelets to the subendothelium [35]. Notably, vWF is also secreted by the endothelial cells and protects circulating factor VIII from the proteolytical degradation [36]. Adhesion leads to cytoskeleton rearrangement and a change in platelet shape resulting in platelet activation [37]. Furthermore, platelets can also be activated directly by thrombin [38], fibrinogen [39], or proinflammatory cytokines such as platelet-activating factor [40]. Activated platelets release a wide range of mediators to facilitate further activation and aggregation of other platelets. P-selectin is translocated from alpha-granules to the platelet surface and binds its ligand P-selectin glycoprotein ligand 1 (PSGL-1) on leukocytes and endothelial cells [41,42]. This interaction promotes rolling of leukocytes and platelets on the activated endothelium. Thus, leukocytes can form a scaffold to fibrin formation. Moreover, P-selectin itself induces fibrin deposition [43]. Adenosine diphosphate (ADP) and thromboxane-A_2_ (TxA_2_) released from dense granules induce vasoconstriction and further platelet activation with increased platelet GP IIb–IIIa expression [37]. Activation of platelet GP IIb–IIIa leads to plug formation by binding vWF and causing fibrinogen deposition on the platelet surface. Activated platelets release coagulation factors, such as factor V and factor VIII, resulting in further fibrin formation [44,45] (Figure 2).

The two main inhibitors of platelet activation and aggregation are the vasoactive nitric oxide (NO) and prostacyclin (PGI_2_) released by the endothelium which work synergistically in platelets [46,47].

In conclusion, physiological haemostasis is regulated by complex interactions between coagulation factors and their regulators, platelets, adhesion molecules, and immune cells and endothelial cells.

## 3. Current Knowledge on the Effects of OSA on Coagulation, Fibrinolysis, and Platelet Activation

In theory, OSA can affect all pathways of the Virchow triad and result in endothelial damage, stasis, and hypercoagulability. Described alterations in the coagulation system induced by OSA are outlined here.

Intermittent hypoxia is one of the primary proposed mechanisms of haemostatic alterations in OSA. After 4 weeks of exposure to intermittent hypoxia, the levels of fibrinogen, factor VIII, and vWF were elevated in an animal model [48]. Intermittent hypoxia modifies the hepatic protein synthesis and aggravates inflammation in the liver which is the major source of coagulant and anticoagulant factors [49]. Increased expression of hypoxia-inducible factor-1 (HIF-1) and transcription factor nuclear-kB (NF-kB) is also mediated by intermittent hypoxia in OSA [50,51,52,53,54] leading to upregulated expression of procoagulant factors, such as TF and factor VIII [55,56,57]. Moreover, PAI-I, VEGF, and NOS genes are also targeted genes of HIF-1 and regulate haemostatic processes [58]. Furthermore, intermittent hypoxia itself increases the production of TF by suppressing the protein C anticoagulant pathway in endothelial cells [59]. Enhanced platelet activity and aggregation were also documented under hypoxic conditions [60,61], and the degree of hypoxia was a significant predictor of platelet activation [62]. In contrast, a recent study demonstrated reduced activation of platelet GP IIb–IIIa under hypoxia resulting in an impaired platelet adhesive function [63].

Increased sympathetic activity caused by intermittent hypoxia and sleep fragmentation has also emerged as an important factor in OSA-associated hypercoagulability [64]. Catecholamines increase the levels of circulating factor V and vWF and directly activate the platelets [65,66]. In the study of Eisensehr et al., elevated epinephrine levels in the morning correlated with increased haemostasis in patients with OSA [67].

Finally, it has previously been described that chronic inflammation itself leads to abnormal haemostasis [68]. Intermittent hypoxia and accompanying oxidative stress may induce the production of proinflammatory cytokines in OSA [69]. Cytokines and chemokines directly and indirectly activate platelets; thus, they release stored proinflammatory substances [70,71]. Cytokines, such as TNFα and IL-1β, also increase the expression of TF [14,72]. The extrinsic pathway can be enhanced in parallel by endothelial dysfunction which is consequently caused by hypoxic and inflammatory processes in OSA [73,74].

The endothelium is the most important factor which regulates coagulation by ensuring adequate blood flow, serving a barrier to subepithelial prothrombotic extracellular matrix components and releasing vasoactive regulatory molecules. It is known that intermittent hypoxaemia leads to endothelial injury contributing to impaired regulation of the coagulation [75]. Endothelin-1 is overexpressed by the endothelial cells in OSA [76] and leads to increased expression of vWF and TF [77,78]. The glycocalyx is a layer covering the endothelium which is composed of glycosaminoglycans, proteoglycans, and plasma proteins [79]. The main regulator of the coagulation cascade, antithrombin III, is bound to heparan sulphate proteoglycans of the glycocalyx [80]. Systemic inflammatory stimuli, hyperglycaemia, oxidised low-density lipoprotein cholesterol (LDL-C), and oxidative stress could damage the glycocalyx, leading to impaired regulation of coagulation [79]. In line with this, increased turnover of hyaluronic acid, an important component of the glycocalyx, has recently been reported in OSA [81]. In addition, endothelial dysfunction also promotes platelet activation and aggregation by increased release of vWF and plasminogen activator inhibitor-1 (PAI-1) and decreased production of NO and PGI_2_ from the endothelial cells [82] (Figure 3).

Clinical studies in patients with OSA supported the findings from basic science and model studies. The levels of TF were elevated in OSA [83,84]. The concentration of TF was directly related to the percentage of time spent with an oxygen saturation < 90% [83] and oxygen desaturation index (ODI) [84] highlighting the role of intermittent hypoxia. Notably, higher plasma levels of TF were also associated with polysomnographic indices of sleep fragmentation in participants without a history of OSA, indicating that sleep disruption itself alters the coagulation system [85]. However, another study did not show a correlation between TF levels and the arousal index in patients with OSA [83]. In the study by Robinson et al., the plasma levels of factor VIIa and factor XIIa were significantly higher in the OSA group compared to controls. However, there was no correlation between these coagulation factors and the severity of OSA [86]. Plasma fibrinogen levels were elevated in patients with OSA in most [87,88,89] but not all studies [90]. In a recent meta-analysis on prothrombotic markers including 2190 participants from 15 studies, patients with OSA had significantly higher plasma fibrinogen levels compared with controls [9]. Wessendorf et al. demonstrated that elevated circulating fibrinogen was associated with the severity of OSA in patients with concomitant history of stroke suggesting a possible link between OSA-associated hypercoagulation and cerebrovascular complications [89] (Appendix A).

Tissue plasminogen activator is released by thrombin, proinflammatory cytokines, and vascular endothelial growth factor (VEGF) from the storage granules in endothelial cells [91]. Markers of systemic inflammation [92], thrombin [86,93], and VEGF levels [94] are increased in OSA, theoretically contributing to high tPA levels. However, the data on the levels of tPA are contradictory, as no difference in tPA levels [95] and activity [96], higher tPA levels [97], and lower tPA activity [95] were consistently reported in OSA. The circadian variation in tPA levels and activity could be a possible explanation for the contradictory results [95]. tPA together with uPA are rapidly cleared by the liver following forming complexes with LDL-receptor like protein [91]; however, it is not clear if this mechanism is altered in OSA. Several drugs, such as steroids, statins, and valproic acid, may induce tPA release. Differences in medication usage of the studied populations could also lead to discrepancies. Only one study investigated uPA levels in OSA and reported lower concentrations [97]. The expression of uPA is induced by oestradiol [98] and survivin [99], the levels of which are decreased in OSA [100,101]. Urokinase is activated by plasmin and kallikrein [102]. The latter was found to show decreased expression in OSA [103]. The soluble levels of uPAR were found unaltered in OSA [104,105]. The expression of uPAR is upregulated by proinflammatory cytokines [28,106,107,108] and TGF-β [109,110]. While inflammation is accelerated [92], reduced levels of TGF-β were reported in OSA [97]. Although suPAR levels in general reflect uPAR expression, it is noteworthy that the cleavage of uPAR may be reduced in OSA due to the low levels of uPA and activated plasmin. Factor XII is a weak activator of plasminogen. Its levels were reported to be increased in OSA [86]. The urinary levels of kallikrein, another weak plasminogen activator were found to be reduced in children with OSA [103] (Appendix A).

The regulator pathways of the coagulation cascade have not been extensively investigated. The levels of APC were comparable between the OSA and non-OSA groups in the study of Takagi et al. [93]. Endothelial protein C receptor (EPCR) and thrombomodulin promote the activation of protein C. Whilst the blood and urinary levels of EPCR were increased in OSA [111], there was no difference in thrombomodulin [93]. Apolipoprotein H inhibits the activation of protein C [112], and their levels were increased in OSA [113]. Whilst antithrombin itself has not been measured in OSA before, the levels of TAT complex were higher in patients with OSA compared to the control group, and there was an association between TAT levels and OSA severity measured by ODI or AHI in some [86,93] but not all studies [114]. Annexin V also has anticoagulant properties as it competes with prothrombin for phosphatidylserine binding sites [115]. Increased frequency of Annexin V+ endothelial cells [116] and Annexin V+ microparticles [117,118] were reported in OSA suggesting that this molecule may serve in a negative feedback mechanism of OSA-related coagulation (Appendix A).

Most studies reported elevated PAI-1 levels in blood samples of patients with adult [95,96,119,120,121,122,123,124] and paediatric [125] OSA. Although the study by Nizankowska-Jedrzejczyk et al. did not find a difference in PAI-1 levels [126], the total number of patients and controls was relatively low (*n* = 38). The levels of PAI-1 are directly related to disease severity [95,119,122,127], emphasising the role of OSA in increased PAI-1 expression. The increased PAI-1 levels in OSA are not surprising, as PAI-1 expression is increased by hypoxaemia [128], systemic inflammation [129], oxidative stress [130], cortisol, and angiotensin II [129]. Another potential mechanism explaining increased PAI-1 levels could be the decreased expression of the klotho in OSA [131]. Klotho is an anti-inflammatory, anti-aging protein, and increased PAI-1 levels were found in klotho deficient mice [132]. However, PAI-1 levels need to be interpreted carefully, as a significant circadian variation of PAI-1 has been described previously [95,124]. This variation is due to both direct control of PAI-1 expression by the circadian genes and diurnal variation of hormones [129]. Interestingly, the variation is larger in patients with OSA than in non-OSA controls [95]. Significantly higher levels of A2AP levels were reported in OSA, and they were related to disease severity [119]. The reason for these changes was not investigated in detail but could be due to the increased IL-6 levels in OSA which induce A2AP formation [133]. The levels of TAFI were reported to be higher in OSA [126]. This molecule is activated by thrombin, plasmin, trypsin, and neutrophil elastase [134]. However, the potential of thrombin to active TAFI is multiplied by thrombomodulin which is unaltered in OSA [93] suggesting that TAFI activation is weak in OSA. The fibrin is stabilised by the complement C3 which has been previously reported to be higher in OSA [135] (Appendix A).

Increased platelet activation and aggregation were described in OSA in several [136,137,138,139] but not all studies [96,140]. Platelet aggregation was higher in severe OSA compared to mild disease [67] and correlated with AHI [136]. GP–Ib is a marker of platelet activity and it is downregulated and internalised during platelet activation [141]. GP–Ib receptor density in platelets was downregulated in OSA indicating increased platelet activation and platelet reactivity; however, GP IIb–IIIa expression did not differ between the OSA and control group [62]. Another marker of platelet activation, P-selectin was measured in higher concentrations in patients with OSA compared to controls in some [86,126,142,143] but not all studies [62,117,144]. The results of studies on the role of vWF in OSA are inconsistent. In some studies, vWF levels were significantly higher in the OSA group compared to controls [83,104]. However, Zamarrón-Sanz et al. did not detect any difference in vWF levels between OSA and controls [145]. Platelet-derived microparticles (PMPs) are generated during platelet activation. PMPs are also suggested to provide an activated surface for the coagulation cascade with 50–100× higher procoagulant activity compared to the activated platelets [146,147]. The levels of PMPs were higher in OSA [148,149] and correlated with disease severity [149] in most but not all studies [139]. Significantly higher platelet counts were detected in OSA [150] and children with OSA [151,152] compared to healthy individuals. Moreover, there was an association between platelet count and disease severity in patients with OSA and manifest cardiovascular disease [153] (Appendix A).

Elevated blood coagulability was confirmed by clinical coagulation tests in OSA. Prothrombin time (PT) is used in clinical practice to evaluate the function of the extrinsic and common coagulation pathways, and activated partial thromboplastin time (aPTT) reflects the abnormalities of the intrinsic pathway [154]. A recent study measured a significantly shorter PT and unchanged aPTT especially in patients with moderate to severe OSA compared to controls, suggesting an activated extrinsic pathway in OSA [150].

OSA is associated with a procoagulant state due to increased levels of coagulation factors, enhanced platelet activation and aggregation, and endothelial dysfunction induced by intermittent hypoxia and inflammatory processes. However, the relationship between OSA and some individual coagulation factors or regulator molecules is controversial. Future randomised controlled studies are warranted to gain a more precise understanding of haemostasis in OSA.

## 4. The Effect of OSA Treatment on Coagulation, Fibrinolysis, and Platelet Activation

Limited data are available on the effects of OSA therapies on haemostatic alterations, and the findings are inconsistent. One month of continuous positive airway pressure (CPAP) therapy failed to decrease the levels of activated factor VIIa, factor XIIa, and factor VIIIa [86]. In contrast, another study demonstrated a significant post-CPAP decrease in 24 h concentrations of factor V, factor VIII, and vWF especially in the nocturnal and morning periods. However, factor VII levels remained unchanged after 2 months of treatment with CPAP [155], yet another study reported a significant decrease in factor VII levels after 6 months of therapy [156]. Whilst some authors found no difference in plasma fibrinogen levels [86,140,155,157], another group detected decreased fibrinogen concentrations in response to CPAP therapy; however, the sample size was small (*n* = 11) in this study [158] (Appendix A).

Neither uPA nor tPA concentrations changed following CPAP treatment [97]. Two studies reported that PAI-1 levels decreased following two weeks [159] and one month [97] of CPAP treatment, respectively. However, another well-designed study did not find any change in PAI-1 levels following two months of CPAP usage [155]. This may suggest that the short-term beneficial effect of CPAP may be reversed by homeostatic factors in the long term. More concordant is the lack of effect of CPAP on the diurnal variability of PAI-1 levels in OSA [121,155]. Treatment with a mandibular advancement device (MAD) did not change PAI-1 concentrations [126]. In contrast, PAI-1 levels significantly decreased following adenotonsillectomy in children with OSA [160]. Finally, PAI-1 concentrations were decreased following sleeve gastrectomy [161] compatible with the fact that PAI-1 partly originates from adipose tissue [129]. TAFI levels significantly decreased following treatment with MAD [126] (Appendix A).

Several studies found a decrease in platelet activation and aggregation after one night or one to three months of CPAP therapy in OSA [136,138,162,163]. However, whilst platelet aggregation was reduced following 90 days of CPAP therapy, there was no difference at 30 days [164]. P-selectin was not influenced either by CPAP [86] or by MAD [126]; however, in another study CPAP resulted in a decrease in P-selectin levels [138]. Most of the studies did not report significant changes in the levels of vWF [83,86,121,159], with one exception [155] which had a cross over design with a 1 month washout period. Significantly decreased levels of vWF were detected 6 months after upper airway surgery [165]. Two weeks of CPAP withdrawal resulted in an elevation in the levels of PMPs [166]; however, the same workgroup also reported conflicting results [167]. Higher body mass index and disease severity of participants in the former study may explain the different results. PT and aPTT were increased following 30 days of CPAP therapy [163] and upper airway surgery [168] (Appendix A).

The effect of OSA therapies on haemostasis is inconclusive. Several studies demonstrated that short- and long-term CPAP therapy had beneficial effects on coagulation system and platelet function in OSA. However, CPAP failed to improve the procoagulant state in OSA in other reports. Further adequately powered randomised controlled studies with higher treatment efficacy and adherence are required to determine the effects of CPAP therapy on the haemostatic alterations.

## 5. Discussion of Major Findings

As outlined above, several pathophysiological consequences of OSA, particularly intermittent hypoxia, sympathetic activity, systemic inflammation, and consequential endothelial dysfunction, may result in a hypercoagulable state, platelet activation, and impaired fibrinolytic capacity. Some data from in-vitro studies and animal models have been confirmed in case control studies in patients with OSA. However, there are also controversial findings, particularly on the effect of OSA treatment on the finding of changes in coagulation or fibrinolysis. Many studies had methodological limitations or were not designed to primarily assess coagulation, and the few randomised controlled trials (RCTs) had a limited sample size. In addition, the severity of OSA and the associated hypoxic burden differed between studies. However, whether the discrepancies between studies are due to the differences in OSA severity and the phenotype or limitations in study design is speculative.

Some findings on hypercoagulability and a disturbed coagulation–fibrinolysis balance have been consistently shown and support the role of hypercoagulability as one of the mechanisms explaining the observed incidence of vascular events in OSA. For instance, elevated levels of PAI-1 are known to increase the risk for myocardial infarction [169,170].

How hypercoagulability and impaired fibrinolysis are affected by comorbidities has not been systematically studied in large sample sizes. Coexistent comorbidities, such as obesity, may lead to systemic inflammation and liver damage that also leads to abnormalities in coagulation [10,171]. This could be an explanation for the lack of change in coagulation factors following CPAP therapy [86]. Of note, most studies assessed short-term effects of CPAP therapy on the coagulation system. It is also unclear whether effective treatment of OSA with CPAP could reduce the cardiovascular risk that is attributable to disturbances in the coagulation system or fibrinolysis.

Limitations in study design that do not allow for causative associations to be established between OSA and changes in coagulation or that do not adequately control for comorbidities or OSA severity or phenotypes are a common problem. However, there are some data from RCTs and meta-analyses that strengthen the level of evidence, e.g., on increased levels of procoagulant and platelet-derived microvesicles or fibrinogen in OSA [166,172].

## 6. Clinical Implications

A procoagulant state in OSA may make OSA patients more susceptible to both venous thromboembolism and thrombus formation on arterial plaques resulting in pulmonary embolism, deep vein thrombosis, acute coronary syndrome, and stroke [1,2,173]. The findings on hypercoagulability and impaired fibrinolysis lead to the hypothesis that OSA both promotes occurrence of vascular events in patients with atherosclerosis due to hypercoagulability and results in more severe end organ damage in case of an ischemic vascular event as a consequence of the impaired clot lysis.

The current knowledge on OSA as risk factor of a procoagulant state should be implemented in treatment recommendations together with comorbidities and other cardiovascular risk factors. In addition, measures of coagulation and fibrinolysis could be used for phenotyping patients and risk assessment.

## 7. Implications for Research

Due to the limitations in study design of previous studies on coagulation and fibrinolysis in OSA and due to relevant confounding factors, such as obesity and comorbidities, conclusions on the causative relationship on OSA and hypercoagulability are somewhat limited, and the quality of evidence needs to be improved. Potential causative associations between OSA and hypercoagulability and the role of comorbidities is a research topic that needs to be further investigated in well-designed studies using different models and designs.

## 8. Summary

In summary, there is evidence that OSA results in elevated levels of fibrinogen and increased platelet activity, promotes platelet adhesion and aggregation, and results in an impaired fibrinolytic capacity. A hypercoagulable state and impaired fibrinolysis promote thrombotic events, and this might be one of the several underlying mechanisms linking OSA with an adverse cardio- and cerebrovascular outcome. Several other pathophysiological consequences of OSA that differ between phenotypes of OSA, clustering of cardiovascular risk factors, and comorbidities might define the vascular risk that OSA induces in an individual. However, hypercoagulability, fibrinolysis, and haemostasis are potential therapy targets that can be influenced either via the coagulation system or the endothelium.

## Figures and Tables

**Figure 1 ijms-22-02834-f001:**
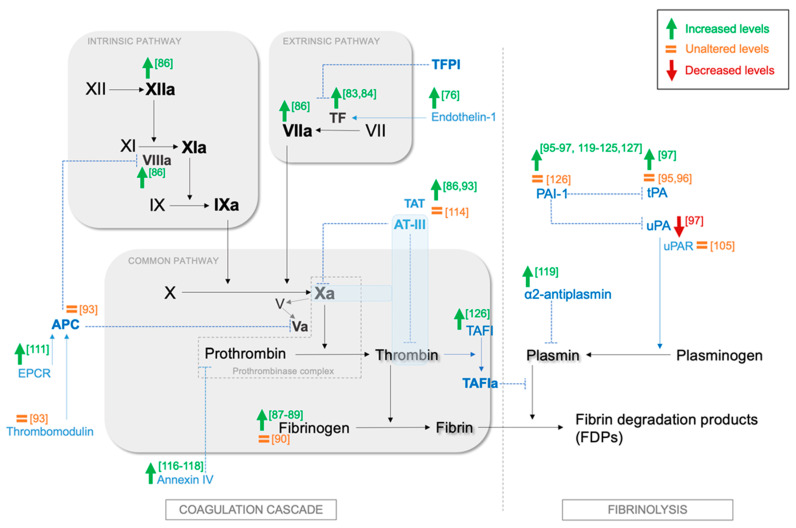
The coagulation cascade and fibrinolysis and how obstructive sleep apnoea (OSA) interacts with them. APC—activated protein C; AT-III—antithrombin III; EPCR—endothelial protein C receptor; IX—factor IX; IXa—activated factor IX; PAI-1—plasminogen activator inhibitor-1; TAFI—thrombin activated fibrinolysis inhibitor; TAFIa—activated thrombin activated fibrinolysis inhibitor; TAT—thrombin-antithrombin complex; TF—tissue factor; TFPI—tissue factor pathway inhibitor; tPA—tissue plasminogen activator; uPA—urokinase plasminogen activator; uPAR—urokinase type plasminogen activator receptor; Va—activated factor V; VIIIa—activated factor VIII; X—factor X; Xa—activated factor X; XI—factor XI; Xia—activated factor XI; XII—Factor XII; XIIa—activated factor XII.

**Figure 2 ijms-22-02834-f002:**
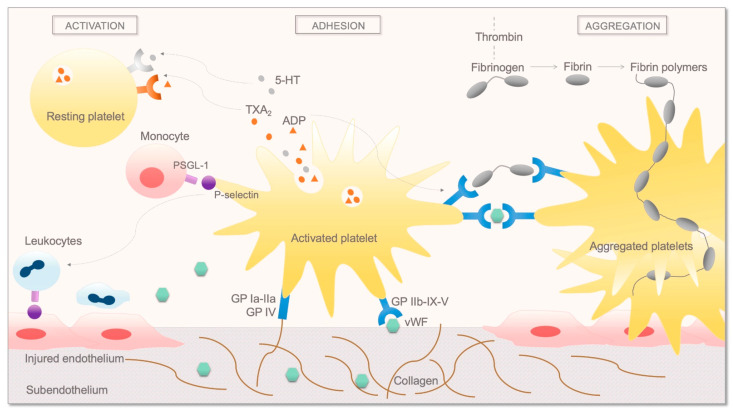
The mechanism of platelet adhesion and aggregation. 5-HT—serotonin; ADP—adenosine diphosphate; GP Ia-IIa—glycoprotein Ia-IIa; GP IIb-IX-V—glycoprotein IIb-IX-V; GP IV—glycoprotein IV; PSGL-1—P-selectin glycoprotein ligand 1; TXA_2_—thromboxane-A2; vWF—von Willebrand factor.

**Figure 3 ijms-22-02834-f003:**
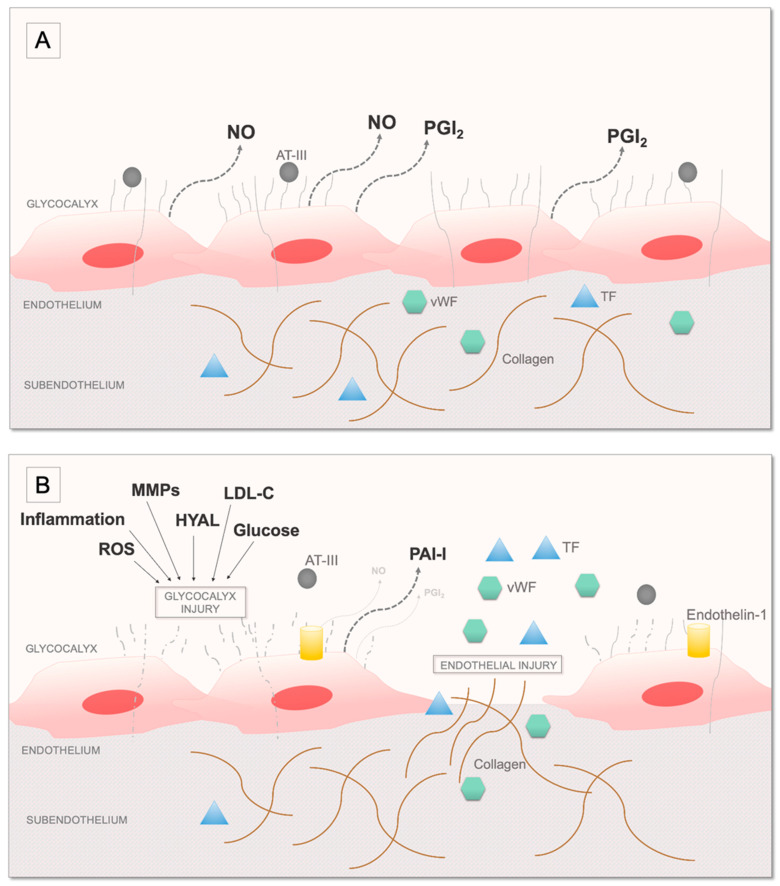
The role of endothelium in regulation of haemostasis (**A**) and endothelial injury (**B**) in OSA. AT-III—antithrombin III; HYAL—hyaluronidase; LDL-C—low-density lipoprotein cholesterol; MMPs—matrix metalloproteinases; NO—nitric oxide; PAI-1—plasminogen activator inhibitor-1; PGI_2_—prostacyclin; ROS—reactive oxygen species; TF—tissue factor; vWF—von Willebrand factor.

## Data Availability

Not applicable.

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
