# Peer review of "Coagulation and Fibrinolysis in Obstructive Sleep Apnoea"

_ijms, 2021, doi:10.3390/ijms22062834_

Round 1
Reviewer 1 Report
Manuscript entitled “Coagulation and fibrinolysis in obstructive sleep apnoea” summarizes available knowledge on the processes associated with coagulation in OSA patients. The undertaken topic is important, particularly taking under consideration the frequent comorbidities in OSA patients, especially DM2 and cardiovascular diseases.
Authors provide and in-depth explanation of the problem and include well presented figures.
For the sentence”In addition, there is growing awareness that there are different phenotypes of OSA based on pathophysiology, symptoms, comorbidities, and long-term cardiovascular consequences.” adding some references would be advisable (eg. doi:10.1016/j.smrv.2016.10.002, doi: 10.1038/s41598-019-56478-9).
It would be interesting to take under consideration the possible involvement of HIF-1alpha as mediator of disused processes in OSA patients as this factor affects transcription of for example:
vascular endothelial growth factor (VEGF), platelet-derived growth factor (PDGF), nitric oxide synthase (NOS) or plasminogen activator inhibitor (PAI). Recently several studies have been published on HIF-1alpha in OSA patients (doi: 10.5664/jcsm.8682, doi: 10.20452/pamw.15104, doi: 10.17305/bjbms.2016.1579, doi: 10.1111/jsr.12995, doi: 10.3390/jcm9051599).
Author Response
Q1: For the sentence ”In addition, there is growing awareness that there are different phenotypes of OSA based on pathophysiology, symptoms, comorbidities, and long-term cardiovascular consequences.” adding some references would be advisable (eg. doi: 10.1016/j.smrv.2016.10.002, doi: 10.1038/s41598-019-56478-9).
A1: Thank you for the recommendations. We added the requested references (Page 1, Line 34).
Q2: It would be interesting to take under consideration the possible involvement of HIF-1alpha as mediator of disused processes in OSA patients as this factor affects transcription of for example: vascular endothelial growth factor (VEGF), platelet-derived growth factor (PDGF), nitric oxide synthase (NOS) or plasminogen activator inhibitor (PAI). Recently several studies have been published on HIF-1alpha in OSA patients (doi: 10.5664/jcsm.8682, doi: 10.20452/pamw.15104, doi: 10.17305/bjbms.2016.1579, doi: 10.1111/jsr.12995, doi: 10.3390/jcm9051599).
A2: Thank you for your valuable suggestions. We expanded the manuscript with this information and the requested references were added (Page 5, Lines 187-191).
Reviewer 2 Report
In this review article, the authors have presented a comprehensive account of the literature on how OSA promote a “prothrombotic” state. The authors have thoroughly covered most of the significant studies in the field. Moreover, they have done a very good job in making clear illustrations. Potential clinical and research implications have been discussed, and the final conclusion summarized the authors’ point of view.
Following are the suggestions on how to further improve this timely review:
1.The authors should indicate in the beginning or end how was the literature search was done, e.g., what key words and which search engine, that will help orient the readers regarding what aspects or time period is covered.
2.The authors should present their perspective at the end of each major heading; what are the major questions that are needed to drive the field forward.
Author Response
Q1: The authors should indicate in the beginning or end how was the literature search was done, e.g., what key words and which search engine, that will help orient the readers regarding what aspects or time period is covered.
A1: Thank you for your suggestion. We included the details of the literature search in the manuscript (Page 2, Lines 49-61).
Q2: The authors should present their perspective at the end of each major heading; what are the major questions that are needed to drive the field forward.
A2: Thank you for raising this important point. We corrected the manuscript and included the main conclusions and future perspectives at the end of the major headings. (Page 5, Lines 175-177; Page 8, Lines 327-331; Page 9, Lines 369-374). This are also discussed separately in detail in headings 5, 6 and 7.